# Immunostimulatory and Antibacterial Effects of *Cannabis sativa* L. Leaves on Broilers

**DOI:** 10.3390/ani14081159

**Published:** 2024-04-11

**Authors:** Mirta Balenović, Zlatko Janječić, Vladimir Savić, Ante Kasap, Maja Popović, Borka Šimpraga, Marijana Sokolović, Dalibor Bedeković, Goran Kiš, Tihomir Zglavnik, Daniel Špoljarić, Fani Krstulović, Irena Listeš, Tajana Amšel Zelenika

**Affiliations:** 1Poultry Center, Croatian Veterinary Institute, Ul. Vjekoslava Heinzela 55, 10000 Zagreb, Croatia; m_balenovic@veinst.hr (M.B.);; 2Department of Animal Nutrition, Faculty of Agriculture, University of Zagreb, Svetošimunska Cesta 25, 10000 Zagreb, Croatia; 3Department of Animal Science and Technology, Faculty of Agriculture, University of Zagreb, Svetošimunska Cesta 25, 10000 Zagreb, Croatia; 4Department of Veterinary Biology, Faculty of Veterinary Medicine, University of Zagreb, Heinzelova ul. 55, 10000 Zagreb, Croatia; 5Regional Veterinary Institute Split, Croatian Veterinary Institute, Poljička Cesta 33, 21000 Split, Croatia

**Keywords:** *Cannabis sativa* L., leaves, immunostimulatory effect, antibacterial effect, broilers

## Abstract

**Simple Summary:**

Food safety, climate change, the emergence of infectious diseases, the ban on the use of antibiotics as growth promoters, and increasingly demanding intensive production are daily challenges for poultry production. A functional immune system is a prerequisite for animal health, and nutrition is one of the modulators of the immune system; therefore, the appropriate balance of nutrients is extremely important for the proper development and maintenance of the immune system of animals. The antimicrobial and immunomodulatory effects of phytobiotics are properties that make their use important as feed additives for poultry. *Cannabis sativa* L. contains many different compounds such as flavonoids, terpenes, and cannabinoids, each with different properties and effects. The effects of *C. sativa* seeds, essential oils, and cakes as feed additives for poultry have already been investigated, but the effect of *C. sativa* L. leaves as a feed additive on immunostimulatory and antibacterial activity has not. The results of this study show that *C. sativa* as a phytogenic additive to animal feed has a favourable antimicrobial and immunomodulatory effect in the production of broiler chickens.

**Abstract:**

The aim of this study was to evaluate the effect of dried *Cannabis sativa* L. leaves as a phytogenic mixture added to broiler feed on CD4^+^ and CD8^+^ T lymphocyte subpopulations, Newcastle disease virus (NDV) antibody titres, and the presence of *E. coli* in faecal samples. The study was conducted on 100 male Ross 308 broilers, divided into four groups of 25 broilers, for a 42-day research period. The groups were housed separately in boxes on a litter of softwood shavings and were fed starter mixture from day 1 to day 21 and finisher mixture from day 22 to day 42. Industrial hemp (*C. sativa*) was grown in the Crkvina area, Croatia (latitude: 45°18′46.8″ N; longitude: 15°31′30″ E). The hemp leaves were manually separated, sun-dried, and ground to a powder. The mixture offered to the control group did not contain cannabis leaves, whereas the three experimental groups received mixtures containing mixed cannabis leaves in a quantity of 10 g/kg, 20 g/kg, or 30 g/kg (E_10, E_20, and E_30, respectively). The mean NDV antibody level was uniform in all study groups until post-vaccination day 14 and increased comparably with time. The percentage of CD4^+^ and CD8^+^ lymphocytes in the peripheral blood subpopulation showed statistically significant differences (*p* < 0.001) in the E_20 group as compared with the control group and both the E_10 and E_30 groups throughout the study period. As the broiler age increased, the CD4^+^-to-CD8^+^ ratios also increased and were statistically significant (*p* < 0.0001) on day 42 in all experimental groups as compared to the control group. Comparing the control group with the experimental groups indicated that the bacterial count was lower in broiler groups having received feed with the addition of 20 g/kg and 30 g/kg *C. sativa* leaves. In conclusion, the *C. sativa* leaves were found to elicit a favourable immunomodulatory effect on cell-mediated and humoral immune responses in broilers via increased CD4^+^ and CD8^+^ lymphocyte subpopulations and higher CD4^+^:CD8^+^ cell ratios, thus indicating enhanced immune function capacity. In addition, *C. sativa* leaves may have complementary effects on the broiler post-vaccination immune response, increase broilers’ resistance to infectious diseases, reduce the effect of stress associated with vaccination, and improve broiler health and welfare.

## 1. Introduction

Broiler breeding is subject to a number of external effects that frequently cause stress and consequently reduce productivity. In an attempt to avoid losses, antibiotics used to be quite frequently applied as agents promoting poultry growth and health while reducing morbidity and mortality. However, the use of antibiotics has resulted in many untoward effects, primarily microorganism resistance and antibiotic residues in meat and eggs, with unfavourable effects on human health [1,2,3]. Therefore, the European Parliament and Council of the European Union enacted the 1831/2003 Regulation stating that antibiotics, with the exception of coccidiostats and histomonostats, could be marketed and used as feed additives only until 31 December 2005. Anticoccidial agents such as ionophore antibiotics have been banned as feed additives since 2013, when medicines in animal feed were limited to therapeutic use based on a veterinary prescription [4]. The worldwide trend of reducing antibiotic use on animal farms has accelerated research on using alternative agents, so-called natural growth promoters, as feed additives (e.g., prebiotics, probiotics, organic acids, enzymes, silicates, plants, spices, etc.) [5,6,7,8,9,10,11]. Phytobiotics are known for their pharmacological effects, thus being used in traditional and alternative human medicine [1]. The chemical structure of the phytobiotic active component varies depending on the harvest season, geographical origin, and part of the plant used (leaves, pulp, etc.). Phytobiotics can be used in solid, dried, or ground form, or as extracts (raw or concentrated), depending on the procedure employed to obtain the active component [12,13]. The antimicrobial and immunomodulatory effects of phytobiotics are important characteristics that enable their use as poultry feed additives [14].

Polysaccharides are very important immunoactive components of phytobiotics. Phytogenic compounds also possess immunomodulatory activity by enhancing immune cell proliferation, cytokine expression, and the antibody titre. Phytobiotic immunogenicity can manifest as enhanced macrophage, lymphocyte, and natural killer cell activity, as well as the stimulation of interferon production, so that plants containing flavonoids and carotenoids can reinforce the immune system [15]. In their review, Rafeeq et al. [16] emphasized that phytobiotics are used to a great extent in the poultry industry for immune system stimulation; for the prevention and control of various bacterial, viral, and protozoal diseases; and as growth promoters. Various plants contain natural polyphenol compounds or flavonoids as their main active components investigated as potential antimicrobial and antioxidative agents. The nutritive composition can directly influence the immune response of broilers because the systemic immune system is strongly managed by the lymphatic tissue associated with the intestine. Thus, phytobiotics administered as feed additives enhance phagocyte activity and the lymphocyte count. Furthermore, the authors stated that some plants elevate the Newcastle disease virus (NDV) antibody titre.

*C. sativa*, known as hemp, is an annual oleaginous plant from the family *Cannabaceae* [17]. It was introduced in Western medicine at the beginning of the 19th century [18]. Hemp is a highly relevant crop with huge social and economic value since it can be used for the production of food, textiles, clothes, biodegradable plastics, paper, dyes, biofuels, animal feed, and lighting oil. The phytochemical components found in hemp are very complex, representing various chemical classes of primary metabolites such as amino acids, fatty acids, alkaloids, and lignans [19,20,21]. Cannabis contains about 600 identified and numerous as yet unidentified but potentially useful compounds. However, of the numerous unique chemical compounds found in this plant, phytocannabinoids are the most important [22]. Ripe cannabis contains hundreds of different compounds that can be divided into the categories of flavonoids, terpenes, and cannabinoids, each of them having different features and effects. Cannabis inflorescences contain cannabinoids (15.77–20.37%), terpenoids (1.28–2.14%), and flavonoids 0.07–0.14%); leaves contain cannabinoids (1.10–2.10%), terpenoids (0.13–0.28%), and flavonoids (0.34–0.44%); stem bark contains sterolides (0.07–0.08%) and triterpenoids (0.05–0.15%); and roots contain sterols (0.06–0.09%) and triterpenoids (0.13–0.24%). These bioactive compounds have been the basis of the traditional medical use of each part of the cannabis plant in various cultures for the thousands of years of its cultivation [23].

More than 20 types of flavonoids have been isolated from the cannabis leaves, flowers, and pollen, including the *O*-glycoside aglycone variants apigenin, luteolin, orientin, vitexin and isovitexin, kaempferol, and quercetin, as well as cannflavin A and cannflavin B, which are specific to *Cannabis* [24]. Numerous studies have investigated the effects of dietary flavonoids on poultry health, growth, and meat quality. Many flavonoids are antioxidants with anti-inflammatory and other properties; thus, phytogenic compounds are considered a potential alternative to antimicrobial agents in poultry [25,26].

The majority of research has been focused on studying the effects of *C. sativa* seed, essential oils, and cake as poultry feed additives [26,27,28,29]. McPartland and Russo [30] reported the flavonoid component in cannabis leaves to be around 1%. However, according to the literature data, there have been no studies investigating the immune and antibacterial effects of *C. sativa* leaves added to animal feed.

Therefore, the aim of this study was to assess the effect of dried *C. sativa* leaves as a phytogenic mixture added to broiler feed on cellular and humoral immunostimulation and antimicrobial activity.

## 2. Materials and Methods

### 2.1. Animal Keeping and Study Design

This study included 100 male Ross 308 broilers divided into four groups of 25 broilers, grown during a 42-day study period. The groups were housed separately in 1.20 × 1.75 m^2^ boxes on a litter of softwood shavings. During the study, broilers were kept under a standard temperature regimen (gradual reduction from 32 °C to 24 °C) and were assigned to a programme of 23 h light (L)/1 h dark (D) (23L:1D cycle). Throughout the study period, the broiler health status, behavioural changes, and deaths were monitored.

Broilers were fed starter mixture from day 1 to day 21 and finisher mixture from day 22 to day 42. Feed and water were available ad libitum. The main meals were formulated so as to meet the nutritive requirements of broilers based on the nutritional requirements of Aviagen [31]. The mixture offered to the control group (C_0) did not contain cannabis leaves, whereas the three experimental groups received mixtures that contained mixed cannabis leaves, i.e., 10 g/kg (group E_10), 20 g/kg (group E_20), or 30 g/kg (group E_30). The ingredients and chemical composition of the mixtures are shown in Table 1.

The chemical composition of the starter and finisher mixtures was determined using the methods recommended by the International Organisation for Standardisation (ISO) for crude fibre (ISO 6865:2000) [32], total fat (ISO 6492:1999) [33], water (ISO 6496:1999) [34], protein (ISO 5983-1:2005; ISO 5983-2:2009) [35,36], starch (ISO 6493:2000) [37], ash (ISO 5984:2022) [38], and minerals (ISO 6491:1998) [39] P, Ca, Na, Cu, Mn, Zn, Fe, Mg, and K (ISO 6869:2000) [40], and a metabolizable energy calculation [41]. The amount of sugar was determined by the Nelson–Somogyi method [42,43].

### 2.2. Industrial Hemp (Cannabis sativa L.) Preparation

Industrial hemp from GEA-COM Ltd. (Budačka Rijeka, Croatia) was grown in the Crkvina area (latitude: 45°18′46.8″ N; longitude: 15°31′30″ E). At the time of harvesting, the plants were four months old; that is, they were in the reproductive phase. The hemp leaves were manually separated, sun-dried, and ground to a powder. Samples of the hemp plants were tested as received and after grinding in the mill to see whether the grinding process had any effect on the final results.

The certified standards for cannabigerol (CBG), cannabinol (CBN), and cannabidiol (CBD) were purchased from HPC Standards GmbH (Cunnersdorf, Germany). Methanol (MeOH) of HPLC-gradient grade was purchased from Thermo Fisher Scientific (Waltham, MA, USA). Analytical-grade formic acid and all other solvents of analytical grade were purchased from Merck (Darmstadt, Germany).

#### 2.2.1. Determination of Nutrient Concentration

After oven drying (60 °C for 72 h), the following parameters were determined: crude fibre, total fat (ISO 6492, 1999) [33], water (ISO 6496, 1999) [34], protein (ISO 5983-1, 2005; ISO 5983-2, 2009) [35,36], ash (ISO 5984, 2022) [38], and minerals (ISO 6491, 1998) [39] P, Ca, Na, Cu, Mn, Zn, Fe, Mg, and K (ISO 6869, 2000) [40] (Table 2).

#### 2.2.2. Determination of Cannabinoid Concentration

After harvesting, the hemp samples were air-dried for 2–3 weeks. Before grinding (in an electric mill), the samples were dried at 50 °C for 12–24 h. For the determination of the total cannabinoids in the hemp samples, we adapted the method described by Jin et al. [23] and Saingam and Sakunpak [44]. In brief, the samples (hemp and feed) were ground using a manual laboratory grinder (Retsch^®^ Z200, Haan, Germany), and 2.0 g of the sample was mixed with 100 mL of methanol. Extraction was performed at room temperature by shaking for 20 min. The solution was then filtered through Whatmann filter paper (black ribbon). Prior to injection in the high-performance liquid chromatography system, the extract was filtered through a 0.45 μm PTFE filter. All samples were prepared and analysed in triplicate.

The profiling of cannabinoids in the extracts was carried out on a Kinetex^®^ C18 column (5 μ C18 4.6 mm, at 30 °C, Phenomenex, Torrance, CA, USA), with a mobile phase composed of 0.1% HCOOH in methanol and isocratic elution for 20 min. The flow rate was 0.7 mL/min, the injection volume was 10 μL, and injections were performed in triplicate for each sample.

The UV/DAD chromatograms were obtained at 210 nm with an acquisition range of the whole spectrum (190–400 nm). Quantitative determinations of selected cannabinoids were performed with a five-point calibration curve made with each standard in the same chromatographic conditions. The calibration curves for cannabigerol (CBG), cannabinol (CBN), and cannabidiol (CBD) were in the range from 1 μg/mL to 50 μg/mL, while for dTHC, it was in the range from 0.1 μg/mL to 10 μg/mL. The quality of this method was confirmed with a determination of the linearity of the calibration curves. The coefficients of determination (R2) for CBG, CBN, and CBD were 0.9994, 0.9995, and 0.9995, respectively. The results of tests for the determination of recovery values for CBG, CBN, and CBD were 96.1%, 98.3%, and 93.5, respectively.

### 2.3. Broiler Vaccination

The Avishield^®^ ND suspension of vaccine against Newcastle disease was applied oculonasally to one-day-old broilers according to the manufacturer’s instructions (Genera Inc., Kalinovica, Rakov Potok, Croatia). One dose of the vaccine contained live lentogenic NDV, strain La Sota, 10^6.0^–10^7.0^ TCID_50_.

### 2.4. Newcastle Disease Antibody Titre Assessment

At the ages of 14, 21, 28, 35, and 42 days, 15 broilers were randomly selected from each study group. Blood (approximately 0.2 mL per broiler) was sampled by brachial wing vein puncture without the addition of anticoagulant. The blood was left at room temperature for 2 h. Serum samples were obtained by centrifugation at 2500× *g* for 15 min at 25 °C, inactivated (56 °C for 20 min), and stored at −20 °C until analysis. The Newcastle disease antibody titre was determined by the haemagglutination inhibition (HI) assay [45] using 4 haemagglutination units of antigen. The same vaccine as described above was used as the homologous antigen in the assay. The antibody titre was expressed as the log^2^ reciprocal.

### 2.5. Flow Cytometry for T Lymphocyte CD4^+^ and CD8^+^ Subpopulations in Peripheral Blood

At the ages of 14, 21, 28, 35, and 42 days, 10 broilers were randomly selected from each study group. Blood (approximately 0.2 mL per broiler) was sampled by brachial wing vein puncture with the addition of heparin sodium as an anticoagulant (Heparin^®^, Belupo Inc., Koprivnica, Croatia). The following monoclonal antibodies from Southern Biotechnology Associates (Birmingham, AL, USA) were used for the identification of T cell subpopulations: R-PE-conjugated mouse anti-chicken CD4 (cat. No. 8210-09) and BIOT-conjugated mouse anti-chicken CD8α (cat. No. 8220-09). All antibodies were of the mouse IgG isotype and were used in a concentration of 1 µg/mL. The leukocyte count in peripheral blood samples (100 mL) was assessed by flow cytometry (Coulter EPICS.XL, Beckman Coulter, Brea, CA, USA). Blood samples were dissolved with phosphate buffer solution (PBS) until a leukocyte concentration of 5.0–9.7 × 10^9^/L. Then, 50 µL of monoclonal antibodies against avian CD^+^ lymphoid markers from Southern Biotechnology Associates (Birmingham, AL, USA) was added to 100 µL of the prepared blood. The samples were tested in triplicate, and 10,000 cells from each sample were analysed on a flow cytometer. The values obtained were presented as the percentage of leukocyte population expression within the cells analysed.

### 2.6. Bacteriology

The presence of Enterobacteriaceae and their count in broiler faeces samples were demonstrated by the ISO 21528-2:2017 method [46]. Pooled faeces samples were tested weekly. Faeces decimal dilutions (from 10^−1^ to 10^−5^) previously prepared according to the ISO 6887-6:2013 and ISO 6887-1:2017 methods [47,48] from the three experimental groups and the control group were inoculated onto blood agar (Columbia agar + 10% sheep blood), Columbia agar (bioMérieux, Craponne, France), Tryptone Bile X-glucuronic chromogenic agar (TBX agar, bioMérieux, France), and Plate Count Agar (PCA, bioMérieux, Craponne, France). The identification of *E. coli* was performed on a Bruker Microflex LT MALDI TOF mass spectrometer (Bruker Daltonics, Bremen, Germany), whereas the presence of the Salmonella genus was demonstrated by use of the ISO 6579-1:2017/A1:2020 method [49].

### 2.7. Statistical Analysis

Statistical analysis was conducted in the R programming environment [50] using several different R packages, such as “tidyverse” [51] for preparation of the data; “descriptr” [52] for descriptive statistical analysis; and “ggpubr” [53] and “rstatix” [54] for recommended testing of the assumptions of the applied statistical model. The statistical analysis was conducted under the 2-way mixed ANOVA model with one repeated-measures factor (time) and one between-groups factor (meal composition). The Shapiro–Wilk test was used for testing the normality of the dependent variables in each cell of the design, Levene’s test for testing the homogeneity of the variances, Box’s M-test for testing the homogeneity of covariances, and Mauchly’s test for testing the assumption of sphericity. The assumptions were met for all of the analysed traits except the assumption of normality for log^2^ NDV titres according to the Shapiro–Wilk test. However, a graphical diagnostic of normality with qqplots revealed that the data were close to a normal distribution, which allowed us to use parametric tests even for this trait. A significant two-way interaction between the above predictors (meal type and time) was determined for all examined traits (except for CD8^+^ T lymphocyte subpopulations), which indicated that the impact of one factor on the outcome variable depended on the level of the other factor (and vice versa). Therefore, in order to decompose a significant two-way interaction into simple main effects, we ran one-way models of the first variable (meal composition) at each level of the second variable (time) and vice versa. Irrespective of the insignificant interaction between the predictors, we used the same approach in the analysis of CD8^+^ T lymphocyte subpopulations for the sake of consistency in the presented results. In the post hoc multiple pairwise comparisons (multiple *t*-tests), in order to control the overall probability of a Type I error (i.e., false positive results), the Bonferroni adjustment was applied. The results of the inferential statistical analysis were presented graphically using “ggplot2” [55].

## 3. Results

### 3.1. Effect of the Addition of C. sativa Leaves on NDV Antibody Titres

There was a statistically significant interaction between the meal composition and post-vaccination time in explaining the variability of the NDV antibody titre in the serum (*F* (12, 144) = 1.97, *p* = 0.031) of the broilers. The incompletely consistent ranking (Figure 1) of the feeding groups at the different post-vaccination times (and vice versa) made it difficult to draw one-way conclusions. However, there were some patterns in the results worth interpretation. Regarding the effect of meal composition, the results depicted on the left side of Figure 1 revealed that E_20 and E_30 had the highest mean NDV antibody titres in their serum and had consistent ranks from the 3rd to 6th weeks after vaccination. However, statistically significant differences in the NDV antibody titre were only determined between E_20 and E_10 and between E_30 and E_10 on day 21 (*p* < 0.05) and day 28 (*p* < 0.001). The determined NDV antibody titres at the different post-vaccination time points within the same feeding groups (right side of Figure 1) expectedly increased with time, with a few exceptions (initial drops observed in groups C_0 and E_10 at the beginning of the trial). The biggest discrepancy in the NDV antibody titre between the examined adjacent time points was determined between the 4th and 5th weeks after vaccination, and this “jump” was consistent across all the examined treatments, including C_0 and E_10. The additional increase between the 5th and 6th weeks was less pronounced, especially in groups fed with a higher share of *C. sativa* in their meals (E_20 and E_30). The NDV antibody titres in the 5th and 6th weeks significantly differed from the NDV antibody titres in the 2nd, 3rd, and 4th weeks after vaccination (with different levels of statistical significance in pairwise comparisons). By taking into account all of the results presented in Figure 1, it can be concluded that groups E_20 and E_30 had similar patterns of changes across the examined time period and that the addition of 20 mg/kg of *C. sativa* in the meal had the most beneficial impact on the immune response properties of vaccinated broilers.

### 3.2. Effect of Adding C. sativa Leaves on Peripheral Blood CD4^+^ and CD8^+^ Lymphocyte Proliferation and CD4^+^:CD8^+^ Ratios

There was a statistically significant interaction between the meal composition and post-vaccination time in explaining the variability in the percentages of the CD4^+^ lymphocyte subpopulation (*F* (4.29, 51.48) = 5.03, *p* = 0.001) and CD8^+^ lymphocyte subpopulation (*F* (7.27, 87.29) = 0.78, *p* = 0.61) in the peripheral blood of broilers, as well as the CD4^+^-to-CD8^+^ ratios (*F* (9108) = 8.91, *p* ≤ 0.0001) determined by flow cytometry.

Regarding the effect of the meal composition on CD4^+^ lymphocyte proliferation, the results shown on the left side of Figure 2 indicate that E_20 showed statistically significant differences (*p* < 0.0001) compared to C_0 over the entire period. Taking into account the time intervals within the same group, the results on the right side of Figure 2 show that a statistically significant difference (*p* < 0.001) was recorded on day 28 compared to day 21 in all groups, which would correspond to the response to NDV vaccination. The percentage of the CD4^+^ lymphocyte subpopulation at different time points within the same feeding group (right side of Figure 3) increased over time (*p* < 0.001), as expected, with the greatest increase in the E_20 group. A very similar pattern of change was also observed in the percentage of the CD8^+^ subpopulation. As with CD4^+^ lymphocyte proliferation, the effect of the meal composition on CD8^+^ lymphocyte proliferation was significantly bigger (*p* < 0.0001) in E_20, in comparison to C_0 (left side of Figure 3). It was also noted that no difference was recorded between groups E_10 and E_30 throughout the period, while a statistically significant difference was recorded between E_10 and E_20 and between E_20 and E_30 (*p* < 0001). The increase between days 35 and 42 was less pronounced in all groups, which was to be expected as lymphocyte proliferation had reached its plateau.

With increasing age of the broilers, the CD4^+^:CD8^+^ ratios increased (left side of Figure 4), and there were statistically significant differences (*p* < 0.0001) in all experimental groups (E_10, E_20, and E_30) compared to the control group (C_0). The differences in the CD4^+^:CD8^+^ ratio in the E_20 group were statistically significant (*p* < 0.0001) when compared to the other groups at 42 days of age. The CD4^+^:CD8^+^ ratio at different time points within the same feeding group (right side of Figure 4) increased over time, but with statistically significant differences (*p* < 0.0001) in the group fed 20 g/kg *C. sativa* in their meals (E_20).

### 3.3. Effect of Adding C. sativa Leaves on E. coli Count in Faeces

The results of the bacteriological examinations of the pooled faeces samples (Table 3) showed statistically significant differences (*p* < 0.0001) in the number of *E. coli* already on the 21st day in E_30 compared to the control group (C_0). Statistically significant differences (*p* < 0.0001) were found on day 35 in all experimental groups (E_10, E_20, and E_30) compared to the control group (C_0), and this trend continued until the end of the study (day 42). The *E. coli* counts in pooled faeces samples at different time points within the same feeding group increased over time, but statistically significant differences (*p* < 0.0001) were observed in group E_20 on days 28 and 35 and in E_30 on day 28. Throughout the study period, bacteria of the genus Salmonella were not detected in any of the pooled faecal samples.

## 4. Discussion

Research into alternative substances for use as feed additives in poultry has intensified since the placement of bans on antibiotics as growth promoters and for poultry health protection [4,6,7,8,9,10,11]. Although the mechanism of phytobiotic action has not yet been fully clarified, numerous studies have been conducted showing their variable efficacy [56].

Our results suggest that various levels of *C. sativa* leaves added to broiler feed influenced the synthesis of NDV antibodies after vaccination. The highest antibody titres were recorded in the E_20 group, which received 20 g/kg of additive, throughout the study period.

The CD4^+^:CD8^+^ lymphocyte ratios obtained by flow cytometry were calculated to determine the relative fluctuation in the number of CD8^+^ cells as compared with CD4^+^ cells. The CD4^+^:CD8^+^ lymphocyte ratio is used as a measure of immune function and response. Low ratios are seen in acute viral diseases, whereas high ratios are associated with enhanced immune function of the body (Figure 4). Our results indicated the CD4^+^:CD8^+^ ratios to have increased with broiler age, and they were quite high at the age of 42 days in all experimental groups. In the E_20 group, differences in the CD4^+^:CD8^+^ ratio were statistically significant (*p* < 0.0001) when compared with other groups at the age of 42 days. As there are no literature data on the effect of *C. sativa* leaves as a poultry feed additive, we believe that flavonoids from the *C. sativa* leaves administered to broilers as a phytogenic additive were among the main constituents that influenced their immune response.

Unlike our study, Mahmoudi et al. [57] reported that hemp seed did not have any major effect on the production of Newcastle disease virus antibodies. In the literature, there are many studies on phytogenic additives such as fennel, lemon balm, pepper, anise, oak, clove, and thyme extracts, which resulted in an increase in the lymphocyte count and Newcastle disease virus antibody titre [6]. Talebi et al. [58] used *Nigella sativa* seed and monitored the immune response to vaccination against Newcastle disease. They found a significant difference in the antibody titre recorded in broilers that received 1% *N. sativa* seed. Besides this increase in the Newcastle disease virus antibody titre, they reported a decrease in the lymphocyte percentage, which was not observed in our study.

T lymphocyte subpopulations in peripheral blood are one of the most relevant indicators of the overall level of immunity. CD4^+^ cells are associated with the major histocompatibility (MHC) class II molecules and act as helper or inflammatory T cells in response to exogenous antigens, whereas CD8^+^ T cells are associated with MHC class I molecules, playing the main role as cytotoxic T cells in response to antigens. The CD4^+^:CD8^+^ ratio is considered a direct indicator of body immunity [59,60]. Lee et al. [61] and Pourhossein et al. [62] demonstrated that feed additives containing phytogenic compounds induced immunomodulatory abilities via immune cell proliferation and antibody titre increase, which is consistent with our results. In his review, Abd El-Ghany [15] also reported on a number of studies investigating the use of phytobiotics, which increased the lymphocyte count and haemagglutination inhibition antibody titre after vaccination against Newcastle disease, while inhibiting *E. coli* growth in the intestine.

In our study, antimicrobial activity was observed at day 35 in the experimental groups that received *C. sativa* leaves as a feed additive, as compared to the control group (C_0). On day 42, 5.53 × 10^6^ cfu/g was recorded in the control group, whereas 3.80 × 10^6^ cfu/g, 3.20 × 10^6^ cfu/g, and 3.13 × 10^6^ cfu/g were measured in the E_10, E_20, and E_30 groups, respectively. Similar results have been reported by Bolukbasi and Erhan [63], They added thyme to feed and recorded lower levels of faecal *E. coli*. In our previous research, we compared the effectiveness of dry marigold, dandelion, and basil flowers, that is, the effectiveness of their flavonoids on the kinetics of auxiliary and cytotoxic T lymphocytes in peripheral blood, and on the number of E. coli colonies in faeces, and we recorded their antimicrobial and immunostimulating effects in laying hens [64].

Many studies listed by Abd El-Hack et al. [14] reported lower *E. coli* counts when using phytogenic feed additives, suggesting that many plants exert favourable effects on intestinal health. The concentration of cannabinoids may vary within the same harvest or fluctuate according to plant ripeness; thus, these features need to be additionally investigated to elucidate the use of industrial hemp.

## 5. Conclusions

*C. sativa* leaves were found to elicit a favourable immunomodulatory effect on the cell-mediated and humoral immune response in broilers via increased CD4^+^ and CD8^+^ lymphocyte subpopulations and higher CD4^+^:CD8^+^ cell ratios, thus indicating enhanced immune function capacity. In addition, *C. sativa* leaves may have complementary effects on the broiler post-vaccination immune response, increase broilers’ resistance to infectious diseases, reduce the effect of stress associated with vaccination, and improve broiler health and welfare.

Research in the field has shown that phytogenic feed additives have favourable antimicrobial and immunomodulatory effects. However, knowledge about their use in poultry nutrition remains inadequate, requiring additional studies and agricultural support through the production of feed additives from locally available plants and spices. The exact mechanism of action of each particular phytobiotic is hard to specify because they occur in plants in the form of various mixtures. This may also be the reason for the differences in their effects among previous studies. In the poultry industry, efforts have long been invested to improve animal welfare. Considering the yet limited knowledge of the effect of *C. sativa* leaves, the impact of the method of drying and storage on their antimicrobial and immunomodulatory features should be explored, and the possible toxicity for the host should be assessed. Attention should also be paid to productivity results, as well as to the meat quality and nutritive values as food for humans.

## Figures and Tables

**Figure 1 animals-14-01159-f001:**
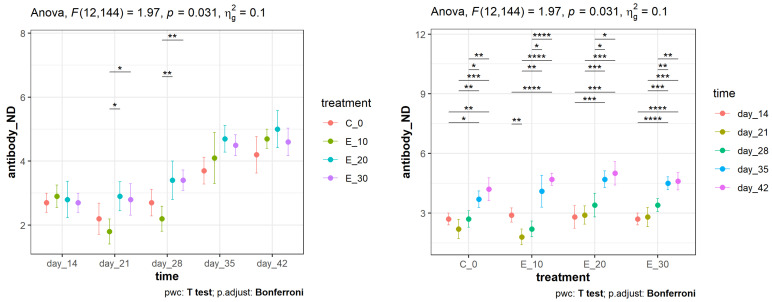
The means (points) and 95% confidence intervals (whiskers) of the Newcastle disease virus antibody titres (log^2^) in the vaccinated broilers fed with different meals and measured at different post-vaccination times. The asterisks on the top of the dot–whisker plot indicate standard levels (* *p* < 0.05, ** *p* < 0.01, *** *p* < 0.001, **** *p* < 0.0001) of statistical significance accounting for the Bonferroni adjustments.

**Figure 2 animals-14-01159-f002:**
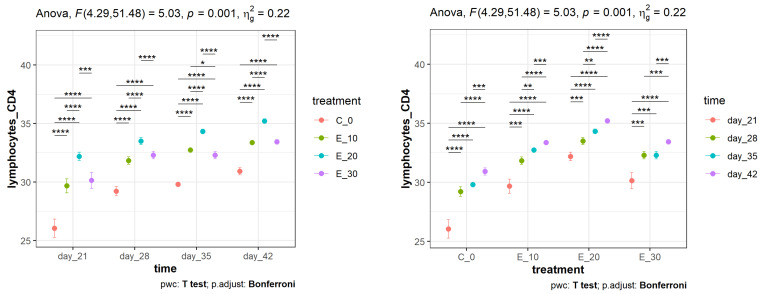
Mean values (points) and 95% confidence intervals (whiskers) of the percentage of the CD4^+^ lymphocyte subpopulation in peripheral blood in broilers fed different diets and measured over a certain time period. Asterisks at the top of the dot plots indicate standard levels (* *p* < 0.05, ** *p* < 0.01, *** *p* < 0.001, **** *p* < 0.0001) of statistical significance taking into account Bonferroni adjustments.

**Figure 3 animals-14-01159-f003:**
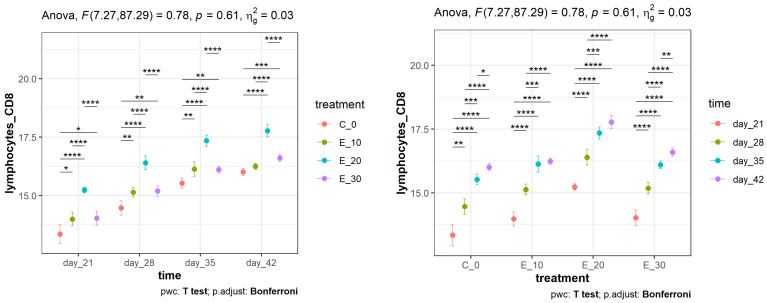
Mean values (points) and 95% confidence intervals (whiskers) of the percentage of the CD8^+^ lymphocyte subpopulation in peripheral blood in broilers fed different diets and measured over a certain time period. Asterisks at the top of the dot plots indicate standard levels (* *p* < 0.05, ** *p* < 0.01, *** *p* < 0.001, **** *p* < 0.0001) of statistical significance taking into account Bonferroni adjustments.

**Figure 4 animals-14-01159-f004:**
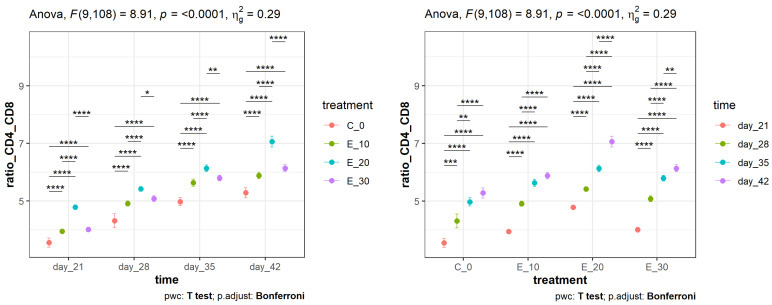
Mean values (points) and 95% confidence intervals (whiskers) of the CD4^+^:CD8^+^ cell ratio determined by flow cytometry in broilers fed different diets and measured over a certain time period. Asterisks at the top of the dot plots indicate standard levels (* *p* < 0.05, ** *p* < 0.01, *** *p* < 0.001, **** *p* < 0.0001) of statistical significance taking into account Bonferroni adjustments.

**Table 1 animals-14-01159-t001:** Mixture composition and calculated composition of standard meals used in the study.

Item	Starter (Day 1–21)	Finisher (Day 22–42)
Ingredient (%)		
Maize	45.20	55.00
Soybean meal (46%)	17.80	9.50
Soybean cake	14.00	14.00
Wheat	11.00	9.50
Sunflower meal (35%)	7.00	7.00
Premix ^1,2^	5.00	5.00
Total	100.00	100.00
Nutrient content analysed		
Crude protein (%)	20.37	17.72
Crude fibre (%)	4.49	4.29
Ash (%)	5.45	3.84
Total fat (%)	5.26	4.31
Water (%)	11.15	12.91
Starch (%)	42.33	46.33
Sugar content (%)	3.96	2.99
Ca (%)	0.98	0.54
P (%)	0.73	0.45
Na (%)	0.196	0.158
Mg (%)	0.20	0.16
K (%)	0.10	0.683
Cu (mg/kg)	13.00	26.00
Mn (mg/kg)	110.00	130.00
Zn (mg/kg)	144.00	146.00
Fe (mg/kg)	362.00	244.00
ME * (MJ/kg)	12.54	12.35

* ME—metabolizable energy. ^1^ Contents per kg feed (Starter): 10,000 IU vitamin A; 2500 IU vitamin D_3_; 30 mg vitamin E; 2 mg vitamin K_3_; 1 mg vitamin B_1_; 3 mg vitamin B_6_; 0.01 mg vitamin B_12_; 25 mg vitamin C; 12 mg Ca-D-pantothenate; 0.5 mg folic acid; 0.2 mg biotin; 750 mg choline chloride. ^2^ Contents per kg feed (Finisher): 10,000 IU vitamin A; 2500 IU vitamin D_3_; 100 mg vitamin E; 2 mg vitamin K_3_; 2.3 mg vitamin B_1_; 2.3 mg vitamin B_6_; 0.01265 mg vitamin B_12_; 100 mg vitamin C; 23 mg Ca-D-pantothenate; 46 mg niacin; 1.725 mg folic acid; 0.0575 mg biotin; 750 mg choline chloride.

**Table 2 animals-14-01159-t002:** Main nutrient contents in original dry matter and determination of cannabinoids in industrial hemp (*C. sativa* L.) leaves.

	Unit	Industrial Hemp Leaves (*C. sativa* L.)
Crude protein	%	18.92
Crude fibre	%	15.75
Ash	%	14.34
Total fat	%	9.69
Water	%	5.26
Minerals		
Ca	%	4.12
P	%	0.43
Na	%	0.017
Mg	%	0.58
K	%	1.61
Cu	mg/kg	15.10
Mn	mg/kg	35.53
Zn	mg/kg	47.96
Fe	mg/kg	186.53
Cannabinoids in hemp leaves		
CBD	µg/kg	242.51
CBG	µg/kg	7.55
CBN	µg/kg	4.55

**Table 3 animals-14-01159-t003:** *Escherichia (E.) coli* count in broiler pooled faecal samples.

Day of Experiment	*E. coli* Count (log CFU g^−1^) × 10^6^x ± SEM			
	C_0	E_10	E_20	E_30
7	4.65 ± 0.06	3.58 ± 0.05 *	2.78 ± 0.05 *	3.78 ± 0.05 *
14	5.15 ± 0.06	5.08 ± 0.01	5.33 ± 0.02	4.48 ± 0.05
21	5.75 ± 0.06	5.02 ± 0.01	5.04 ± 0.02 **	4.16 ± 0.02 **
28	5.58 ± 0.05	4.08 ± 0.05 **	3.78 ± 0.05 *^,^**	3.58 ± 0.05 *^,^**
35	5.50 ± 0.04	3.93 ± 0.05 *	3.18 ± 0.05 *^,^**	3.18 ± 0.05 *^,^**
42	5.53 ± 0.03	3.80 ± 0.05 *	3.20 ± 0.07 *	3.13 ± 0.08 *

* Statistically significantly higher value (*p* < 0.0001) compared to the value determined in the control group in the same period. ** Statistically significantly higher value (*p* < 0.0001) in relation to the day of sampling within the same group.

## Data Availability

Data are contained within the article.

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
