# Peer review of "Immunostimulatory and Antibacterial Effects of Cannabis sativa L. Leaves on Broilers"

_animals, 2024, doi:10.3390/ani14081159_

Round 1

Reviewer 1 Report

Comments and Suggestions for Authors

All comments and suggestions are presented in the pdf format.

Author Response

Dear Reviewer,

Please find our comments in the attached file.

Authors

Reviewer 2 Report

Comments and Suggestions for Authors

Comments on the Quality of English Language

The English language is of adequate quality.

Author Response

(The authors gave the same response as above.)

Reviewer 3 Report

Comments and Suggestions for Authors

This is a very timely and interesting study.  It is the first study I've seen using hemp leaves, as much more work has been done with hemp seed meal.  I'm not an immunologist so I won't comment much on the lymphocyte ratios or immune response.  Although, the higher titers with the hemp diets does offer some intriguing speculation.  There is much we do not know about phytogenic feed additives and there potential to address a variety of issues.  The results related to the E. coli counts are quite interesting.  How these lower E. coli counts might affect overall litter quality and the bacterial load and birds going to the processing plant would be interesting as well.

I am less knowledgeable about the immunomodulatory aspects but the favorable antimicrobial effects that you present certainly warrant further research.  As you mention, how the cannabinoid concentration will vary due to conditions will be critical to determine.  In addition, the unknowns you describe...possible toxicity to the birds, meat quality, nutritional value, productivity parameters (weight gain, FCR, livability, welfare aspects) all must be determined.  While you have a well-written and valuable paper, it seems to have generated more questions than answers (but that is not necessarily a bad thing).  The area of phytogenic feed compounds is in need of much additional research and the poultry industry is in desperate need of this research with the removal of antibiotics from feed.

Line 71 - replace "organ" with "organic"

Line 172 - what stage of growth (or age) were the hemp plants at harvest (vegetative, reproductive)?

Author Response

(The authors gave the same response as above.)
